# GSK-3β in Dendritic Cells Exerts Opposite Functions in Regulating Cross-Priming and Memory CD8 T Cell Responses Independent of β-Catenin

**DOI:** 10.3390/vaccines12091037

**Published:** 2024-09-10

**Authors:** Chunmei Fu, Jie Wang, Tianle Ma, Congcong Yin, Li Zhou, Björn E. Clausen, Qing-Sheng Mi, Aimin Jiang

**Affiliations:** 1Center for Cutaneous Biology and Immunology, Department of Dermatology, Henry Ford Health, Detroit, MI 48202, USA; cfu1@hfhs.org (C.F.); jwang5@hfhs.org (J.W.); cyin1@hfhs.org (C.Y.); lzhou1@hfhs.org (L.Z.); 2Immunology Program, Henry Ford Cancer Institute, Henry Ford Health, Detroit, MI 48202, USA; 3Department of Medicine, College of Human Medicine, Michigan State University, East Lansing, MI 48824, USA; 4Department of Computer Science and Engineering, School of Engineering and Computer Science, Oakland University, Rochester, MI 48309, USA; tianlema@oakland.edu; 5Department of Internal Medicine, Henry Ford Health, Detroit, MI 48202, USA; 6Institute for Molecular Medicine, Paul Klein Center for Immune Intervention, University Medical Center of the Johannes Gutenberg-University Mainz, Langenbeckstrasse 1, 55131 Mainz, Germany; bclausen@uni-mainz.de

**Keywords:** dendritic cells, GSK-3β, β-catenin, CD8 T cell immunity, memory responses

## Abstract

GSK-3β plays a critical role in regulating the Wnt/β-catenin signaling pathway, and manipulating GSK-3β in dendritic cells (DCs) has been shown to improve the antitumor efficacy of DC vaccines. Since the inhibition of GSK-3β leads to the activation of β-catenin, we hypothesize that blocking GSK-3β in DCs negatively regulates DC-mediated CD8 T cell immunity and antitumor immunity. Using CD11c-GSK-3β^−/−^ conditional knockout mice in which GSK-3β is genetically deleted in CD11c-expressing DCs, we surprisingly found that the deletion of GSK-3β in DCs resulted in increased antitumor immunity, which contradicted our initial expectation of reduced antitumor immunity due to the presumed upregulation of β-catenin in DCs. Indeed, we found by both Western blot and flow cytometry that the deletion of GSK-3β in DCs did not lead to augmented expression of β-catenin protein, suggesting that GSK-3β exerts its function independent of β-catenin. Supporting this notion, our single-cell RNA sequencing (scRNA-seq) analysis revealed that GSK-3β-deficient DCs exhibited distinct gene expression patterns with minimally overlapping differentially expressed genes (DEGs) compared to DCs with activated β-catenin. This suggests that the deletion of GSK-3β in DCs is unlikely to lead to upregulation of β-catenin at the transcriptional level. Consistent with enhanced antitumor immunity, we also found that CD11c-GSK-3β^−/−^ mice exhibited significantly augmented cross-priming of antigen-specific CD8 T cells following DC-targeted vaccines. We further found that the deletion of GSK-3β in DCs completely abrogated memory CD8 T cell responses, suggesting that GSK-3β in DCs also plays a negative role in regulating the differentiation and/or maintenance of memory CD8 T cells. scRNA-seq analysis further revealed that although the deletion of GSK-3β in DCs positively regulated transcriptional programs for effector differentiation and function of primed antigen-specific CD8 T cells in CD11c-GSK-3β^−/−^ mice during the priming phase, it resulted in significantly reduced antigen-specific memory CD8 T cells, consistent with diminished memory responses. Taken together, our data demonstrate that GSK-3β in DCs has opposite functions in regulating cross-priming and memory CD8 T cell responses, and GSK-3β exerts its functions independent of its regulation of β-catenin. These novel insights suggest that targeting GSK-3β in cancer immunotherapies must consider its dual role in CD8 T cell responses.

## 1. Introduction

β-catenin, a central component of the canonical Wnt signaling pathway, has emerged as a key regulator of dendritic cell (DC) function [1,2]. Recent studies by our group and others have shown that tumors, including melanoma, induce the upregulation/activation of β-catenin in DCs, thereby suppressing the antitumor CD8 T cell response [3,4,5,6,7]. The serine/threonine kinase Glycogen synthase kinase-3 (GSK-3), a key upstream kinase for β-catenin, plays a critical role in various cellular functions including differentiation, apoptosis, tumor growth, immune responses, cell invasion, and metastasis [8]. Aberrant GSK-3 activity has been associated with many human diseases, including cancers [9]. Recent studies have also demonstrated the role of GSK-3 in antitumor immunity by regulating the functions of immune cells including DCs, especially in the tumor microenvironment [10], thus highlighting GSK-3 as a potential therapeutic target for cancer therapy. Indeed, many GSK-3 inhibitors have been developed, and some are currently being tested in clinical trials [8,10,11].

The most studied form of GSK-3 is GSK-3β, which is constitutively active in resting DCs, where it is thought to phosphorylate β-catenin, thereby marking it for proteasomal degradation [12]. However, the function of GSK-3β in DCs is still not fully understood. The activation of GSK-3β has been shown to promote apoptosis in DCs, thereby limiting DC-mediated immune responses [13], whereas the inhibition of GSK-3β enhances DC maturation and immune function [12]. Although the inhibition of GSK-3 is often associated with upregulated/activated β-catenin [12,14,15], the role of upregulated/activated β-catenin in mediating the effects of GSK-3 has not been thoroughly investigated.

The inhibition of GSK-3β has also been shown to improve the efficacy of DC vaccines, through the regulation of 2,3-dioxygenase (IDO), which is consistent with the negative role of GSK-3 in immune responses [15]. However, a recent study in human DCs showed that constitutively active GSK-3β resulted in improved DC development, maturation, and antitumor efficacy of DC vaccines [16], suggesting that GSK-3β may play a positive role in antitumor immunity. Since these studies mainly relied on pharmacological inhibition of GSK-3β, we aimed to determine the function of GSK-3β in the regulation of CD8 T cell responses by inhibiting GSK-3β genetically.

In this report, using mice with a CD11c-specific deletion of GSK-3β (CD11c-GSK-3β^−/−^ mice), we surprisingly found that the lack of GSK-3β did not upregulate β-catenin expression at the transcriptional or protein level in CD11c^+^ DCs. CD11c-GSK-3β^−/−^ mice exhibited significantly augmented cross-priming and antitumor immunity, opposite to the phenotypes observed with a CD11c-specific stabilization of β-catenin (CD11c- β-catenin^active^ mice [3]). Despite augmented cross-priming, CD11c-GSK-3β^−/−^ mice completely abrogated memory CD8 T cell responses upon DC-targeted vaccinations, suggesting that GSK-3β in DCs also plays a positive role in regulating the differentiation and/or maintenance of memory CD8 T cells. scRNA-seq analysis revealed that the deletion of GSK-3β in DCs positively regulated transcriptional programs for the effector differentiation and function of primed antigen-specific CD8 T cells in CD11c-GSK-3β^−/−^ mice, but negatively regulated transcriptional programs for memory cells, resulting in a significant loss of CD8 T cell populations with memory gene signatures. In conclusion, our data establish that GSK-3β plays opposite roles in DCs in regulating cross-priming and memory CD8 T cell responses, and that GSK-3β exerts its functions independently of β-catenin.

## 2. Results

### 2.1. CD11c-GSK-3β^−/−^ Mice Exhibit Augmented Antitumor Immunity Compared to GSK-3β^Flox/Flox^ Mice

Our previous work has demonstrated that β-catenin activation in CD11c^+^ DCs suppresses antitumor immunity, as CD11c-β-catenin^active^ mice in which the GSK-3β phosphorylation site (exon3) of β-catenin was deleted in CD11c^+^ DCs displayed significantly enhanced tumor growth [3]. Since GSK-3β-mediated phosphorylation of β-catenin marks β-catenin for subsequent ubiquitylation and degradation, we hypothesized that CD11c-specific deletion of GSK-3β would phenocopy β-catenin activation, leading to impaired antitumor immunity. To test this, we monitored tumor growth in wild type (WT, GSK-3β^Flox/Flox^) and CD11c-GSK-3β^−/−^ mice inoculated with B16F10 melanoma cells. Unexpectedly, CD11c-GSK-3β^−/−^ mice exhibited significantly slower tumor growth (Figure 1A), leading to substantially smaller tumor sizes compared to WT mice (Figure 1B,C). These findings disprove our initial hypothesis and suggest an opposite role for GSK-3β in CD11c^+^ DCs compared to β-catenin.

### 2.2. Distinct Gene Expression Profiles of DCs from CD11c-GSK-3β^−/−^ and CD11c-β-catenin^active^ Mice

To better understand how GSK-3β deficiency in DCs affects global gene expression and how it differs from β-catenin activation in DCs, we performed scRNA-seq on sorted splenic DCs from CD11c-GSK-3β^−/−^ and CD11c-β-catenin^active^ mice, as well as their WT littermates. Splenic CD11c^+^ DCs were clustered based on gene expression using an unsupervised inference analysis. A total of 13 distinct clusters were identified and visualized by uniform manifold approximation and projection for dimension reduction (UMAP) algorithm [17] (Figure 2A). Analysis of differentially expressed genes (DEGs) of key DC markers (Figure 2B) and top DEGs (Figure 2C) revealed a clear separation of 13 cell populations, including conventional DCs (cDCs), plasmacytoid DCs (pDCs), and monocyte-derived DCs (MoDCs).

Among the distinct DC populations, clusters 0, 6, 7, 8, and 9 (Figure 2A) represent cDC2 cells with high expression of cDC2 marker genes such as *Sirpa*, *Sirpb1a*, *Itgam* (*CD11b*), *Clec4a1*, *Clec4a2*, and *Apobec1* (Figure 2B). In addition, cluster 9 cells are distinguished by unique expression of G2/M cell cycle genes such as *Cenpe*, *CCnb2*, *Cenpf*, *Cdkn3*, *Birc5*, *Cdca3*, and *Ube2c* [18], while cluster 7 cells express G1/S cell cycle genes like *Mcm5*, *Cdca7*, *Mcm2*, *Mcm3*, and *Hells* [19] (Figure 2C), indicating their potential proliferative capability. Clusters 1 and 10 represent pDCs with high expression of the marker genes *Siglech*, *Bst2*, *Irf7*, and *Pacsin1*. Cluster 2 consists of MoDCs with prominent expression of the monocyte genes *Cd209a and Ccr2.* Cluster 3 cells emerge as IFN-producing killer dendritic cells (IKDCs) [20,21] with a unique NK gene signature (*Klra9*, *Klrb1c*, *Prf1*, *Ncr1*, *Nkg7*, and *Gzmb*) as their top DEGs. Cluster 4 cells are characterized by the expression of *Fscn1*, *Cd63*, and *Ccr7*, suggesting that they are migratory DCs [22]. Cluster 5 are cDC1 cells with high expression of the cDC1 signature genes *Cadm1*, *Xcr1*, and *Clec9a*, while cluster 11 cells express both cDC2 and pDC genes, suggesting they might be the transitional DCs [23,24]. Cluster 12 cells are distinguished by the fact that all of their top DEGs, except *S1pr1*, are B cell markers (*Fcmr*, *Fcer2a*, *Pax5*, *Cd19*, *Ebf1*, and *Cd79a*), resembling recently reported B-pDCs [25].

Interestingly, our scRNA-seq analysis revealed intriguing differences in DC populations of the 13 UMAP clusters between GSK-3β^−/−^ and β-catenin^active^ DCs (Figure 2D). pDCs (cluster 1) were predominant among GSK-3β^−/−^ DCs compared to WT cells, while this phenomenon was not evident in β-catenin^active^ DCs (Figure 2D). Conversely, IKDCs (cluster 5) were significantly reduced in GSK-3β^−/−^ DCs compared to their WT counterparts, while their frequency remained comparable in β-catenin^active^ DCs versus WT DCs. Similarly, cDC1 (cluster 5) were slightly reduced in GSK-3β^−/−^ DCs compared to their WT counterparts, whereas β-catenin^active^ DCs exhibited an increase in cluster 5 cells relative to WT (Figure 2D). Taken together, these data indicated that GSK-3β and β-catenin might involve in DC development in a distinct manner.

Analysis of DGEs of overall DCs revealed almost entirely distinct expression patterns between GSK-3β^−/−^ DCs and β-catenin^active^ DCs comparing to their WT counterparts (Figure 2E,F), with minimal overlap of DEGs between β-catenin^active^ and GSK-3β^−/−^ DCs (0.8% downregulated genes and 3.9% upregulated genes, Figure 2E). Specifically, among the genes significantly downregulated in GSK-3β^−/−^ DCs, most were either upregulated or showed no significant change in β-catenin^active^ DC (Figure 2F). Similarly, among the genes significantly upregulated GSK-3β^−/−^ DCs, most showed no significant change in β-catenin^active^ DC, with only a few consistently upregulated (Figure 2F). Furthermore, Gene Ontology (GO) analysis confirm that β-catenin^active^ and GSK-3β^−/−^ DCs exhibit differentially regulated pathways (Figure 2G). For example, while GSK-3β^−/−^ DCs exhibited top downregulated pathways such as “cellular response to heat”, “response to unfolded protein” and “regulation of mitotic spindle organization”, β-catenin^active^ DCs exhibit downregulation pathways associated with “NF-κB signaling”, “positive regulation of programmed cell death”, and “positive regulation of transcription by RNA polymerase II”. These results strongly suggested that GSK-3β deletion and β-catenin upregulation/activation in DCs have distinct effects on gene regulation patterns.

**In addition, we examined the expression of immune checkpoint molecules previously reported to be differentially expressed in** β**-catenin active DCs** [26]. DCs from CD11c-GSK-3β^−/−^ mice and CD11c-β-catenin^active^ mice displayed different expression profiles of these inhibitory molecules, including PD-L1/L2, Tim-3, and Lag3 (Appendix A). Specifically, while Tim-3 (*Havcr2*) was upregulated in multiple clusters of β-catenin^active^ DCs, GSK-3β^−/−^ DCs exhibited decreased Tim-3 expression compared to WT DCs (Appendix A). Taken together, these data suggest that GSK-3β may regulate DCs independent of β-catenin.

### 2.3. Lack of GSK-3β Does Not Lead to Accumulation of β-catenin in DCs and GSK-3β^−/−^ DCs Exhibit Different Phenotypes than β-catenin^active^ DCs

The opposite antitumor phenotypes of CD11c-GSK-3β^−/−^ and CD11c-β-catenin^active^ mice, along with distinct gene expression patterns of their DCs, prompted us to ask whether GSK-3β deletion in DCs leads to the accumulation of β-catenin. We first examined the expression of β-catenin in GSK-3β^−/−^ DCs. Splenic cDCs were isolated from WT and CD11c-GSK-3β^−/−^ mice, and examined by Western blot for their protein expression. As expected, GSK-3β^−/−^ DCs lost the expression of GSK-3β but not GSK-3α (Figure 3A), confirming the specific GSK-3β deletion in CD11c^+^ DCs. Expression of GSK-3β was not altered in either β-catenin^active^ or β-catenin^−/−^ DCs (Figure 3A). Surprisingly, no elevated expression of β-catenin was observed in GSK-3β^−/−^ DCs compared to WT DCs, in contrast to the substantial accumulation of β-catenin in β-catenin^active^ DCs (Figure 3A,B), suggesting that the deletion of GSK-3β does not result in the accumulation/upregulation of β-catenin in DCs.

We further examined β-catenin expression of GSK-3β^−/−^ DCs by flow cytometry of splenocytes using the gating strategy depicted in Appendix A. Consistent with our Western blot analysis, CD11c^+^Bst2^−^ splenic cDCs from WT and GSK-3β^−/−^ mice exhibited similar expression of β-catenin (Figure 3C,D), further confirming that the deletion of GSK-3β does not lead to the accumulation/upregulation of β-catenin in DCs.

Based on our previous observation that the activation of β-catenin leads to different expression profiles of immune checkpoint molecules in β-catenin^active^ versus WT splenic cDC1 [26], we examined the expression of these inhibitory molecules on WT and GSK-3β^−/−^ cDC1s. In contrast to β-catenin^active^ cDC1s that exhibit increased expression of Tim-3 and PD-L2 and reduced PD-L1 [26], the deletion of GSK-3β in cDC1s did not significantly alter the expression of Tim-3, PD-L1, PD-L2, and Lag3 (Appendix A–D). These data indicate that GSK-3β^−/−^ DCs exhibited a different phenotype in immune checkpoint expression from β-catenin^active^ DCs, further suggesting that the deletion of GSK-3β unlikely regulates DC function through the accumulation/upregulation of β-catenin. Taken together, our data demonstrate that the deletion of GSK-3β in DCs does not upregulate/activate β-catenin and leads to a different gene expression profile than the activation of β-catenin.

### 2.4. Deletion of GSK-3β in DCs Augments DC Vaccine-Induced Cross-Priming of Antigen-Specific CD8 T Cells

We have previously shown that β-catenin in DCs negatively regulates antigen-specific CD8 T cell responses with a DC-targeting vaccine model using anti-DEC-205-OVA [3,4]. We thus asked how deletion of GSK-3β in DCs regulated antigen-specific CD8 T cell responses. WT and CD11c-GSK-3β^−/−^ mice were adoptively transferred with OVA-specific Thy1.1^+^ CD8 T (OTI) cells, and immunized with anti-DEC-205-OVA plus adjuvant CpG. Intriguingly, the percentages of Thy1.1^+^ OTI out of total CD8 T cells, as well as differentiated IFN-γ^+^Thy1.1^+^ OTI effectors out of total OTI CD8 T cells, were significantly augmented in CD11c-GSK-3β^−/−^ mice compared to WT for both spleen and LNs (Figure 4A,B). Furthermore, CD11c-GSK-3β^−/−^ mice exhibited higher frequencies of IFN-γ^+^TNF-α ^+^ and IFN-γ^+^IL-2^+^ polyfunctional effectors than WT mice (Appendix A). Taken together, these data suggest that the deletion of GSK-3β in DCs amplifies OVA-specific CD8 T cell proliferation and differentiation into effector cells.

### 2.5. Deletion of GSK-3β in DCs Led to Diminished Memory Responses Despite Augmented Cross-Priming

As CD11c-GSK-3β^−/−^ mice displayed elevated effector differentiation of primed OTI CD8 T cells, we next examined CD8 T cell memory responses after DC-targeted vaccination. WT and CD11c-GSK-3β^−/−^ mice were immunized with anti-DEC-205-OVA plus CpG following the adoptive transfer of naïve CFSE-labeled Thy1.1^+^ OTI cells, as before, and then challenged at day 21 with OVA protein to assess memory CD8 T cell responses. Despite augmented cross-priming of OTI cells in CD11c-GSK-3β^−/−^ mice (Figure 4A,B), Thy1.1^+^ OTI cell numbers were greatly reduced in CD11c-GSK-3β^−/−^ mice, suggesting that the deletion of GSK-3β in DCs results in dampened CD8 recall responses (Figure 4C). In conclusion, the deletion of GSK-3β in DCs negatively regulates memory CD8 T cell responses despite playing a positive role in augmenting cross-priming.

### 2.6. scRNA-Seq Analysis of OTI CD8 T Cells Primed in WT and CD11c-GSK-3β ^−/−^ Mice

To better understand how the deletion of GSK-3β in DCs affects the differentiation of OVA-specific OTI CD8 T cells, we performed scRNA-seq on sorted Thy1.1^+^ OTI CD8 T cells from WT and CD11c-GSK-3β^−/−^ mice on day 4 and day 10 after immunization with anti-DEC-205-OVA plus CpG. OTI CD8 T cells were clustered based on gene expression using an unsupervised inference analysis, identifying a total of nine clusters (Figure 5A,B). Analysis of top DEGs revealed a clear separation of the nine distinct OTI T cell populations following immunization (Figure 5C).

These clusters exhibit distinct gene expression profiles: for example, cluster 6 represents effector OT1 cells with high expression of *Gzma*, *Gzmb*, and *Ifng*, whereas cluster 7 consists of “interferon-stimulated” CD8 T cells distinguished by interferon gene markers including *Isg15*, *Ifit1*, *Ifit3*, and *Igs20* [27]. Cluster 8 represents a rare “B-T” cluster characterized by elevated expression of B cell markers such as *Cd79a*, *Mef2c*, *Iglc3*, *Ebf1*, and *Ms4a1*(*CD20*), as reported in previous studies [28,29,30]. Notably, clusters 2–4 are distinguished by their top DEGs being the G2/M cell cycle genes *Ube2c*, *Cenpe*, *CCnb2*, *Birc5*, *Cdca8*, *Cks1b*, and Hmgn2, and/or the G1/S cell cycle genes *Mcm5*, *Uhrf1*, *Lig1*, *Cdca7*, *Mcm3*, *Hells*, and *Rfc3*, suggesting that these clusters consist of proliferative OTI CD8 T cells (Figure 5C).

At day 4 post immunization, primed OTI CD8 T cells dissociated into four major populations (clusters 0 and 2–4), with a slight increase in proliferative clusters (clusters 2–4) in CD11c-GSK-3β^−/−^ mice compared to WT mice (Figure 5D). By day 10, cluster 0 and cluster 1 became dominant, with the frequency of cluster 1 (effector memory precursors) in particular being significantly higher in CD11c-GSK-3β^−/−^ compared to WT mice (Figure 5D). At this time point, OTI cells from CD11c-GSK-3β^−/−^ mice also exhibited slightly higher proportions of proliferative clusters (clusters 2–4) and the rare B-T cluster (cluster 8) compared to OTI cells from WT mice (Figure 5D). On the other hand, OTI cells from CD11c-GSK-3β^−/−^ mice exhibited much lower percentages of memory CD8 T cells (cluster 5, 4.0% vs. 10.2%) and effector CD8 T cells (cluster 6, 3.2% vs. 6.2%), as well as a slight reduction in interferon-stimulated CD8 T cells (cluster 7) (Figure 5D). Notably, the percentages of memory CD8 T cells (cluster 5) and effector CD8 T cells (cluster 6) were much higher in OTI cells from CD11c-GSK-3β^−/−^ mice than that of WT mice at day 4 after immunization (8.5% vs. 4.8% for cluster 5; 6.0% vs. 4.3% for cluster 6) (Figure 5D).

Considering that OTI cells from CD11c-GSK-3β^−/−^ mice have only half of the cells in clusters 5 and 6 compared to that from WT mice at day 10 after immunization despite having more cells in clusters 5 and 6 at day 4 (Figure 5D), we focused on these clusters and visualized their top DEGs as violin plots. As shown in Figure 5E, cluster 6 exhibited high expression of *Cx3Cr1*, *Cxcr6*, *Gzma/b*, *Ccr2*, *Zeb2*, and *IFN*, mostly effector markers but some memory markers. For cluster 5, effector genes such as *Gzma/b*, *IFN*, and *Prf1* were down, but expression of memory markers was highest including *Tcf7*, *Il7r*, *Sell*, *Tox*, *Eomes*, *Tbx21*, and *Ctla4*. Interestingly, cluster 5 cells also expressed high *Tnf*, *Ccr7*, and *Lef1*, suggesting that cluster 5 might represent memory cell populations that have retained effector characteristics [31,32]. Thus, a substantially reduced cluster 5 for OTI cells primed in CD11c-GSK-3β^−/−^ mice at day 10 likely led to a marked reduction in memory OTI cells, despite a higher number of cluster 6 effector OT1 cells at day 4 (Figure 5D), consistent with the observed augmented cross-priming but diminished memory responses (Figure 4). Further analysis of effector and memory scores confirmed this hypothesis. Consistent with increased effector T cell priming, OTI CD8 T cells in CD11c-GSK-3β^−/−^ mice had an overall higher effector score compared to OTI cells in WT mice at day 4 (Figure 5F,G). However, at day 10, the effector score of OTI cells in CD11c-GSK-3β^−/−^ mice became lower than that of OTI cells in WT mice (Figure 5F,G). Similarly, the memory score of OTI cells in CD11c-GSK-3β^−/−^ mice was slightly lower than that of OTI cells in WT mice (Appendix A).

Next, we performed a signaling pathway analysis against the MSigDB_Hallmark database on OTI cells primed in WT and CD11c-GSK-3β^−/−^ mice, focusing on the DEGs (*p* < 0.05) in OTI cells primed in CD11c-GSK-3β^−/−^ mice. As shown in Figure 5H, OTI cells primed in CD11c-GSK-3β^−/−^ mice exhibited both significantly upregulated and downregulated pathways at day 4 and day 10 (Figure 5H). Intriguingly, effector and memory function-related TNF-alpha signaling, IL-2/STAT5 Signaling, and IFN-gamma responses were all downregulated in OTI cells primed in CD11c-GSK-3β^−/−^ mice, suggesting that GSK-3β^−/−^ DCs likely interact with primed OTI CD8 T cells to regulate their effector and memory function. Taken together, our scRNA-seq findings indicate that the deletion of GSK-3β in DCs negatively regulated the generation and/or maintenance of memory antigen-specific CD8 T cells, despite elevated effector function during priming.

## 3. Discussion

Here, we reported that that the deletion of GSK-3β in DCs exerts opposite functions in regulating DC vaccine-induced cross-priming and memory responses of antigen-specific CD8 T cells, using a genetic approach to specifically delete GSK-3β in CD11c^+^ DCs compared to β-catenin activation.

GSK-3β phosphorylation targets β-catenin for ubiquitination and subsequent degradation. Unexpectedly, the deletion of GSK-3β in DCs (GSK-3β^−/−^ DCs) did not lead to augmented β-catenin expression and differed in transcriptional regulation from that of activation of β-catenin in DCs, suggesting that GSK-3β exerts its function in DCs independent of β-catenin. Indeed, CD11c-GSK-3β^−/−^ mice exhibited significantly augmented cross-priming and antitumor immunity, phenotypes opposite to those observed with β-catenin active mice (CD11c-β-catenin^active^ mice). However, despite augmented cross-priming, memory CD8 T cell responses are completely abolished in CD11c-GSK-3β^−/−^ mice upon DC-targeted vaccination, suggesting that GSK-3β in DCs also plays a positive role in regulating memory CD8 T cell differentiation and/or maintenance. Further scRNA-seq analysis reveals that CD11c-GSK-3β^−/−^ mice positively regulated transcriptional programs for effector differentiation and function of primed antigen-specific CD8 T cells at day 4 following immunization. However, by day 10, these mice exhibited a substantial loss of populations with memory gene signatures, consistent with diminished memory responses. Thus, our data have demonstrated that GSK-3β plays opposite functions in DCs regulating cross-priming and memory CD8 T cell responses, and GSK-3β exerts its functions independently of β-catenin.

GSK-3, comprising two highly homologous isoforms, GSK-3α and GSK-3β, was initially identified as a regulator of glycogen metabolism [33]. Recent studies have uncovered a broader role for GSK-3, particularly GSK-3β, in regulating innate and adaptive immune responses [34]. In DCs, GSK-3β plays a critical role in determining cytokine production such as IL-10 during inflammation, and regulates DC activation and differentiation [15,35,36]. The inhibition of GSK-3β, often achieved through small molecule inhibitors or RNAi, has been shown to regulate DC differentiation, maturation [12,37], and to mediate CCR7-dependent stimulation of survival [13]. As GSK-3-mediated phosphorylation of β-catenin leads to its degradation, it is not surprising that upregulated/activated β-catenin often accompanies the inhibition of GSK-3 [12,14,15]. However, whether β-catenin activation mediates the effects of GSK-3β remains unclear.

To determine whether the deletion (inhibition) of GSK-3β in DCs regulates DC function through β-catenin, we employed a genetic approach to specifically delete GSK-3β in CD11c^+^ DCs using CD11c-GSK-3β^−/−^ mice. Unexpectedly, the deletion of GSK-3β did not lead to β-catenin upregulation in DCs, as confirmed by both Western blot and flow cytometry (Figure 3). Furthermore, scRNA-seq data revealed substantial differences in gene expression patterns between GSK-3β^−/−^ DCs and β-catenin^active^ DCs, with minimally overlapping DEGs (Figure 2), suggesting that GSK-3β deletion does not result in upregulation of β-catenin at the protein or transcriptional levels. Thus, our finding that the deletion of GSK-3β in DCs does not lead to β-catenin upregulation is different from previous studies implicating that the inhibition of GSK-3β upregulates β-catenin in DCs [12,14,15]. One possible explanation for this discrepancy is that different approaches are used to inhibit/delete GSK-3β: while our genetic approach specifically deletes GSK-3β without affecting other closely related kinases, including GSK-3α (Figure 3), these studies use GSK-3 inhibitors, which block GSK-3α and other related kinases in addition to GSK-3β, suggesting that the inhibition of GSK-3α and other related kinases might account for the observed upregulation of β-catenin.

Further supporting the notion that the deletion of GSK-3β does not upregulate β-catenin in DCs, we also find that CD11c-GSK-3β^−/−^ mice exhibit enhanced antitumor immunity (Figure 1), and upon DC vaccination, augmented cross-priming of antigen-specific CD8 T cells (Figure 4), phenotypes opposite to diminished cross-priming and antitumor immunity of CD11c-β-catenin^active^ mice [3]. Our findings align with a previous report that the inhibition of GSK-3β in DCs improved DC vaccine efficacy [15]. However, it is worth noting that a recent study with human DCs has demonstrated that constitutively active GSK-3 enhances DC function in tumor microenvironment [16], suggesting that GSK-3 in DCs might play either a positive or a negative role in regulating antitumor immunity. Indeed, despite augmented cross-priming, memory CD8 T cell responses were completely lost in CD11c-GSK-3β^−/−^ mice after recall at 21 days post DC vaccination (Figure 4), suggesting that GSK-3β in DCs likely plays an additional role in positively regulating DC vaccine-induced CD8 T cell responses. The phenotype of augmented cross-priming but diminished memory responses of antigen-specific CD8 T cells by CD11c-GSK-3β^−/−^ mice is similar to the phenotype of CD11c-β-catenin^−/−^ mice [4]. In CD11c-β-catenin^−/−^ mice, we have shown that β-catenin exerts both positive and negative regulation on vaccine-induced CD8 T cell responses through IL-10, as the activation of β-catenin in DCs led to upregulated IL-10 production [4]. GSK-3 in DCs has also been implicated in regulating IL-10, although studies have shown that the inhibition of GSK-3 in DCs led to increased IL-10 production [16,36,37,38,39,40], suggesting that GSK-3β unlikely achieves its opposite functions in cross-priming and memory responses through IL-10.

To understand how GSK-3β in DCs exerts opposite functions in cross-priming and memory responses of antigen-specific CD8 T cells, we performed scRNA-seq on primed OTI cells from WT and CD11c-GSK-3β^−/−^ mice on days 4 and 10 after immunization. The scRNA-seq data revealed nine distinct clusters with varied expression of effector and memory markers, confirming the heterogeneity of tumor antigen-specific CD8 T cell responses after DC vaccination [31,32]. Consistent with augmented cross-priming, OTI CD8 T cells in CD11c-GSK-3β^−/−^ mice exhibited an overall higher effector index score compared to OTI cells in WT mice at day 4 (Figure 5). However, by day 10, these OTI cells exhibited a lower effector index score as well as memory index score than OTI cells in WT mice (Figure 5 and Appendix A), suggesting that primed OTI cells from CD11c-GSK-3β^−/−^ mice could not maintain the function of effector/memory CD8 T cells. More strikingly, cluster 5, the memory cell population with highest expression of memory gene markers, was substantially reduced in CD11c-GSK-3β^−/−^ mice at day 10 (Figure 5). The substantial loss of memory CD8 T cell population at day 10 after immunization suggests that the differentiation into and/or maintenance of memory cells from primed CD8 T cells are impaired in CD11c-GSK-3β^−/−^ mice. Thus, our scRNA-seq analysis indicates that the deletion of GSK-3β in DCs leads to not only the reduction in memory CD8 T cell population, but also the function of memory cells. Supporting this notion, we have observed that memory CD8 T cells were almost completely lost at 21 days post immunization, as memory responses in CD11c-GSK-3β^−/−^ mice were abrogated after recall at day 21 (Figure 4). Future studies are warranted to elucidate the mechanisms contributing to the loss of memory CD8 T cells in CD11c-GSK-3β^−/−^ mice.

How does the deletion of GSK-3β in DCs exert opposite function in cross-priming and memory responses of antigen-specific CD8 T cells? We performed signaling pathway analysis on DEGs identified by scRNA-seq data. Interestingly, effector and memory function-related TNF-alpha signaling, IL-2/STAT5 Signaling, and IFN-gamma responses were all downregulated in OTI cells primed in CD11c-GSK-3β^−/−^ mice, suggesting that GSK-3β^−/−^ DCs might interact with antigen-specific OTI CD8 T cells to regulate their effector and memory function through these pathways. Future studies are warranted to further elucidate how individual pathways are employed by GSK-3β to exert its functions on CD8 T cells. Taken together, our data support a model in which the deletion of GSK-3β in DCs regulates transcriptional programs in primed antigen-specific CD8 T cells to augment their effector differentiation and function during the priming phase (at day 4) but impair the generation and/or maintenance of memory CD8 T cells, resulting in diminished memory responses (Figure 6).

GSK-3 inhibitors have emerged as promising agents in cancer immunotherapy due to their ability to modulate immune responses [8,10,11], particularly by suppressing the function of immune cells such as dendritic cells (DCs) and T cells. Recent studies have shown that GSK-3β upregulates the expression of inhibitory receptors LAG-3 and PD-1 in T cells [41], highlighting its potential as a therapeutic target in cancer immunotherapy. However, the regulation of these immune checkpoint molecules appears to be cell-type-specific, as Lag-3 expression was not affected in GSK-3β^−/−^ DCs (Supplemental Appendix A). While the inhibition of GSK-3β has been demonstrated to improve the efficacy of DC vaccines [15], GSK-3β can also play a positive role in boosting antitumor efficacy of DC vaccines [16].

In this report, we showed that the deletion of GSK-3β in DCs led to enhanced antitumor immunity, and augmented DC vaccine-induced cross-priming, supporting the antitumor effects of GSK-3β inhibition. Despite enhanced cross-priming of antigen-specific CD8 T cells upon DC-targeted vaccination, however, memory CD8 T cells were completely lost in CD11c-GSK-3β^−/−^ mice, suggesting that GSK-3β inhibition may ultimately suppress durable antitumor CD8 T cell immunity essential for long-term tumor control. These findings highlight the importance of considering GSK-3β ‘s dual role in CD8 T cell responses when targeting GSK-3 in cancer immunotherapies. For combination therapies with DC vaccines, our data suggest that GSK-3 inhibitors may achieve better antitumor efficacy if applied selectively during the priming phase.

## 4. Conclusions

Here, we report that the deletion of GSK-3β in DCs does not upregulate β-catenin expression and induces a unique gene expression pattern. Notably, CD11c-GSK-3β^−/−^ mice display significantly enhanced cross-priming and antitumor immunity, but completely abrogated memory CD8 T cell responses following DC-targeted vaccination. Further scRNA-seq analysis revealed that the deletion of GSK-3β in DCs positively regulated transcriptional programs for effector differentiation and function of primed antigen-specific CD8 T cells, but negatively regulated the generation and/or maintenance of memory CD8 T cells. Thus, our data suggest that GSK-3β deletion in DCs plays opposing roles in cross-priming and memory CD8 T cell responses, likely independent of β-catenin. These findings underscore the need to consider GSK-3β’s dual role in CD8 T cell responses when targeting GSK-3 in cancer immunotherapies.

## 5. Methods

Mice and treatment. CD11c-GSK-3β^−/−^ (CD11c-Cre^+^ GSK-3β^Flox/Flox^) mice were generated by crossing GSK-3β^Flox/Flox^ with CD11c-Cre transgenic mice, which were generously provided by Dr. J. R. Woodgett (Ontario Cancer Institute, Toronto, Canada). CD11c-β-catenin^active^ (CD11c-Cre^+^β-catenin^Exon3/Exon3^) and CD11c-β-catenin^−/−^ (CD11c-Cre^+^β-catenin^Flox/Flox^) mice were generated and maintained as previously described [3,4]. CD8 TCR transgenic Thy1.1^+^ Rag1^−/−^ OTI mice have been backcrossed >20 generations [42]. C57BL/6 mice were purchased from Charles River. Primary and recalled memory responses were examined as previously described [3]. For DC-targeted vaccinations, mice were injected intravenously (i.v.) with 10 mg of anti-DEC-205-OVA, with CpG (100 mg) subcutaneously (s.c) as adjuvant. All procedures on animals followed protocols approved by the Institutional Animal Care and Use Committee at Henry Ford Health.

Antibodies and reagents. Anti-Thy1.1 magnetic microbeads were obtained from Miltenyi Biotec (Auburn, CA, USA). Antibodies to Thy1.1, TCR Vα_2_, Vβ_5.1/5.2_, CD8α, Siglec-H, Bst-2, CD11b, CD62L, CD44, CD11c, MHC class II I-A^b^, MHC class I H-2K^b^, CTLA-4, Tim-3, PD-L1, PD-L2, Lag-3, PD-1, CCR7, IFN-γ, TNF-α, IL-2, and Granzyme B were purchased from Biolegend Inc. (San Diego, CA, USA). Staining for surface and intracellular antigen expression was performed as previously described [1]. In brief, cells from spleen or pooled draining LN were stimulated for 5 h in vitro with OTI_257–264_ peptide (4 μg/mL, AnaSpec, Fremont, CA, USA) in the presence of Brefeldin A (BFA, 5 μg/mL, Biolegend) and stained for cell surface protein expression followed by fixation and permeabilization, and staining for intracytoplasmic staining antigens such as TNF-αOTI CD8 T cells were labeled with 5-(6)-carboxyfluorescein diacetate succinimidyl diester (CFSE), following flow cytometry for purity check before transfer. We used a Celesta™ (BD Biosciences, Franklin Lakes, NJ, USA) or NovoCyte Quanteon (Agilent Technology, Santa Clara, CA, USA), and data were analyzed using the FlowJo^®^ 10.10 software (Tree Star, Ashland, OR, USA).

In vivo cross-priming assays. In vivo cross-priming assays were carried out as we have reported previously [3]. Briefly, 0.5–1 × 10^6^ CFSE-labeled naïve OTI Thy1.1^+^ CD8^+^ T cells were injected i.v. by tail vein in 200 μL PBS, at −1, 0, or 1 days after immunization with anti-DEC-205-OVA plus adjuvant, followed by in vitro restimulation and subsequent analysis by flow cytometry.

scRNA-seq of splenic DCs and primed OTI CD8 T cells. For splenic DCs, spleen cells were stained with DC markers (CD11c, Siglec-H, Bst2, MHCII), and both CD11c^high^Bst2^−^ cDCs and CD11c^intermediate^Bst2^+^ pDCs were sorted. For isolation of adoptively transferred OTI Thy1.1 CD8^+^ T cells, spleen cells from immunized mice were first enriched with anti-Thy1.1 magnetic microbeads and sorted. scRNA-seq libraries on sorted cells were generated using the 10X Genomics Chromium Single Cell 3′ Reagent Kit (v2 Chemistry) and Chromium Single Cell Controller as described in our previous published study [43]. Briefly, sorted cells were loaded into each reaction for gel bead-in-emulsion (GEM) generation and cell barcoding. Reverse transcription (53 °C for 45 min, 85 °C for 5 min, and a 4 °C hold) of the GEM (GEM-RT) was performed, with a Veriti™ 96-Well Fast Thermal Cycler (Applied Biosystems, Waltham, MA, USA). cDNA amplification was performed after GEM-RT cleanup with Dynabeads MyOne Silane (Thermo Fisher Scientific, Waltham, MA, USA) with the following program: 98 °C for 3 min, 98 °C for 15 s, 67 °C for 20 s, 72 °C for 1 min, then a 12 cycle repeat at 72 °C for 1 min followed by a 4 °C hold. Amplified cDNA was cleaned up with SPRIselect Reagent Kit (Beckman Coulter, Brea, CA, USA) that was followed by a library construction procedure, including fragmentation, end repair, adapter ligation, and library amplification. Library quality control was carried out with an Agilent 2100 Bioanalyzer (Agilent, Santa Clara, CA, USA). Libraries were sequenced on an Illumina HiSeq4000 (Illumina, San Diego, CA, USA) using a paired-end flow cell: Read 1, 26 cycles; i7 index, 8 cycles; Read 2, 98 cycles.

Data analysis for scRNA-seq. Sequenced reads from scRNAseq libraries were demultiplexed, aligned to the mm10 mouse reference, barcode processed, and Unique Molecular Identifier (UMI) counted using the 10X Genomics Cell Ranger (v2.0.1) pipeline [44], as described previously [45]. The R Seurat package (Version 5.0) [46,47] was utilized to analyze the datasets. Combined samples were analyzed with principle component analysis (PCA). Quality control metrics employed are as follows. We employed two strategies to identify potential doublets. Firstly, cells expressing both X and Y chromosome genes (Kdm5d, Eif2s3y, Gm29650, Uty, and Ddx3y) were excluded from the dataset. Secondly, we excluded cells expressing uncharacteristically high numbers of genes (>4000), as well as low-quality cells based on a low number of genes detected (<300) and/or having high mitochondrial genetic content (>5%). Additionally, uninteresting sources of variation within the data were removed. Genes removed include ribosomal structural proteins (as identified by gene ontology term GO:0003735 and the Ribosomal Protein Gene (RPG) database [48]), noncoding rRNAs, Hbb, and genes not expressed in ≥3 cells.

We employed a global-scaling normalization method “LogNormalize” in Seurat to normalize gene expression measurements of each cell by the total expression, multiplied by a factor of 10,000, followed by log-transformation. Highly variable genes in each data analysis were identified, and the intersecting top 1000 genes of each dataset were used for clustering and subsequent downstream analyses. The number of principal components (PCs) used for cell clustering was determined by the Jackstraw method, and the number of canonical correlation components (CCs) used was determined by manual inspection of scree plots. After identifying the number of PCs and CCs to be included for downstream analyses (first 30 PCs), a graph-based clustering approach implemented in Seurat was used to iteratively cluster cells into groups, based on similarities of those components among cells. The uniform manifold approximation and projection (UMAP) method was utilized to visualize resulting clusters. The FindAllMarkers (Version 5) function in Seurat was then used to identify DEGs between clusters with a Bonferroni adjustment of *p* value < 0.05 as a statistical significance threshold. DEGs of two groups were identified with fold change greater than 1.2, with an adjusted *p* value less than 0.05, and Gene Set Enrichment Analysis (GSEA) using Python was used for signaling pathways analysis.

Tumor cell lines and treatment of tumor-bearing mice. B16F10 melanoma cells were inoculated by s.c. injection. Tumors were measured every other day after they become palpable (about 3–5 days), and tumor sizes were calculated as (0.5 × short length × long length^2^). For treatments, tumor-bearing mice were treated when the tumor was about 3 mm–6 mm at about 4–7 days after inoculation. Increased tumor volume was calculated as tumor volume on the day of measurement minus the tumor volume at the time of immunization. Mice were euthanized when tumors exceeded 20 mm in any one dimension or when signs of illness were observed.

Statistical analysis. The statistical significance was evaluated with Excel or GraphPad Prism 9 using a two-tailed unpaired Student’s *t*-test and a linear mixed model with ANOVA for tumor growth. *p* values less than 0.05 were considered significant.

## Figures and Tables

**Figure 1 vaccines-12-01037-f001:**
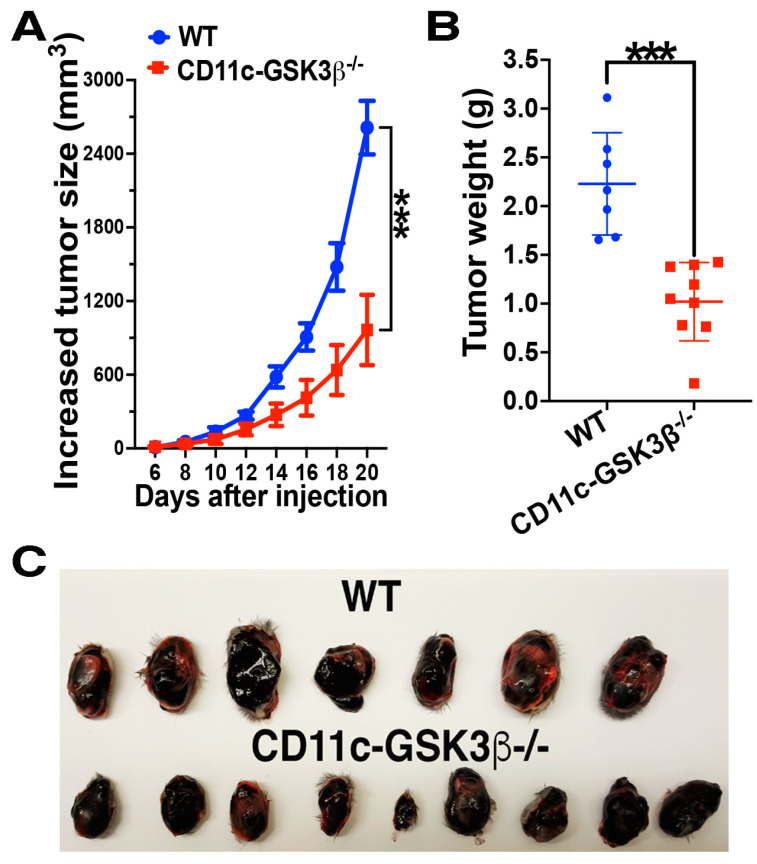
Deletion of GSK-3β in DCs led to augmented antitumor immunity in CD11c-GSK-3β^−/−^ mice. WT and CD11c-GSK-3β^−/−^ mice (*n* = 7–9) were inoculated with B16F10 melanoma cells, and tumor sizes were monitored. (**A**,**B**) CD11c-GSK-3β^−/−^ mice exhibited reduced tumor growth compared to WT mice. Tumor sizes from the day of treatment are shown in (**A**) and tumor weights at the end of the experiment (day 20) are shown in (**B**). A linear mixed model (Lme4) was fitted to the data in (**A**), and ANOVA for the fitted linear mixed model was then performed to determine the difference between groups. Student’s *t*-tests were used for (**B**). *** *p* < 0.001. (**C**) Photo of the tumors at the day 20 after tumor inoculation. Data are representative of two experiments.

**Figure 2 vaccines-12-01037-f002:**
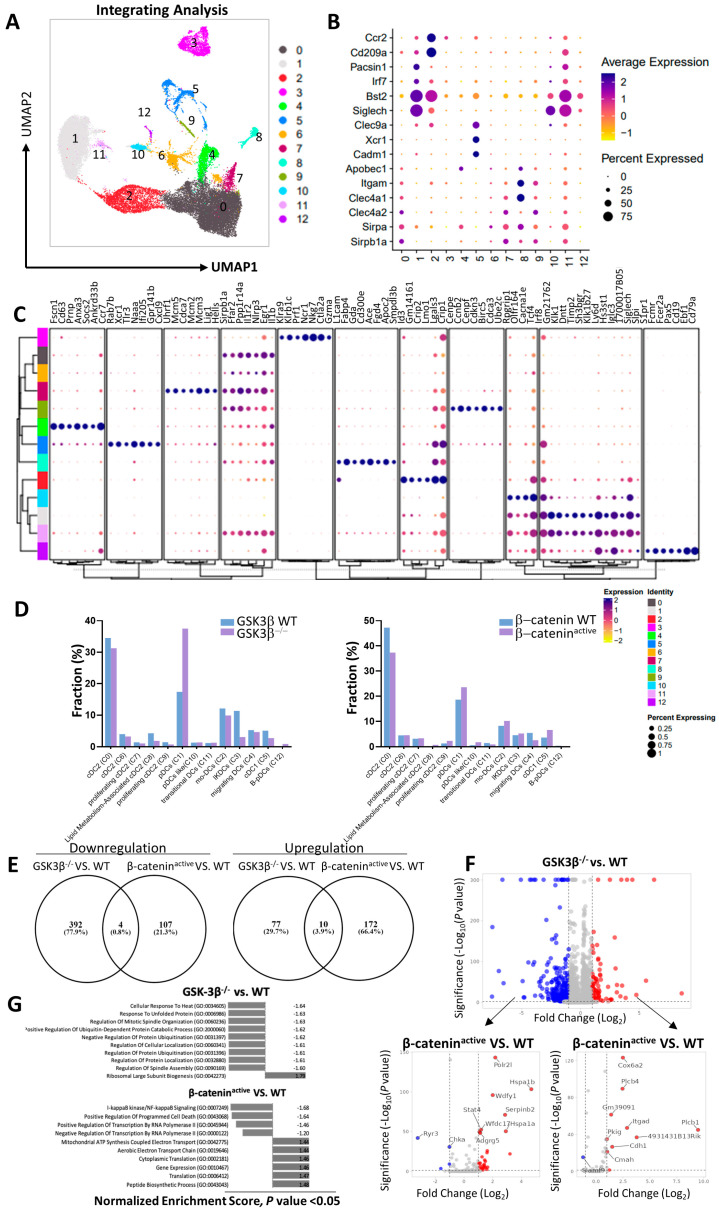
GSK-3β^−/−^ DCs exhibited different expression profiles from β-catenin^active^ DCs by scRNA-seq. DCs sorted from spleens of WT (GSK-3β^Flox/Flox^) and CD11c-GSK-3β^−/−^ mice, or from WT (β-catenin ^Exon3/Exon3^) and CD11c-β-catenin^active^ (CD11c-Cre β-catenin^Exon3/Exon3^), were subjected to scRNA-seq as described. (**A**) Uniform manifold approximation and projection (UMAP) dimensionality reduction mapping analysis of single-cell gene expression of integrated WT (GSK-3β^Flox/Flox^) and GSK-3β^−/−^ DCs, and WT (β-catenin^Exon3/Exon3^) and β-catenin^active^ DCs. Each dot represents one single cell. A total of 13 clusters were identified and color-coded as indicated. (**B**) Bubble plots showing the expression of key markers for pDC, cDC1, cDC2, and MoDCs cells among 13 UMAP clusters. The sizes of dots represent the percentages expressed; the color of dot represents the average expression. (**C**) Bubble plots depicting expression of top DEGs for UMAP clusters shown in (**A**). (**D**) Distribution of cells from WT/GSK-3β^Flox/Flox^ and GSK-3β^−/−^ (left), or WT/β-catenin^Exon3/Exon3^ and β-catenin^active^ DCs (right) within each of the 13 clusters as depicted in (**A**). (**E**) Venn plot showing the overlap of downregulated DEGs (left) and upregulated DEGs (right) in GSK-3β^−/−^ DCs versus WT/GSK-3β^Flox/Flox^ DCs (GSK-3β^−/−^ vs. WT), and β-catenin^active^ and WT/β-catenin^Exon3/Exon3^ DCs (β-catenin^active^ vs. WT). (**F**) Volcano plot visualizing expression of DEGs in GSK-3β^−/−^ and WT/GSK-3β^Flox/Flox^ DCs, and their expression pattern in β-catenin^active^ and WT/β-catenin^Exon3/Exon3^ DCs. DEGs in GSK-3β^−/−^ DCs versus WT/GSK-3β^Flox/Flox^ DCs are shown in volcano plot (upper), and expression of downregulated DEGs (lower left) and upregulated DEGs (lower right) in β-catenin^active^ and WT/β-catenin^Exon3/Exon3^ DCs are analyzed and shown in volcano plots. (**G**) GO enrichment analysis identifies top regulated biological process pathways in in GSK-3β^−/−^ DCs vs. WT/GSK-3β^Flox/Flox^ DCs (upper), and β-catenin^active^ vs. WT/β-catenin^Exon3/Exon3^ DCs (lower).

**Figure 3 vaccines-12-01037-f003:**
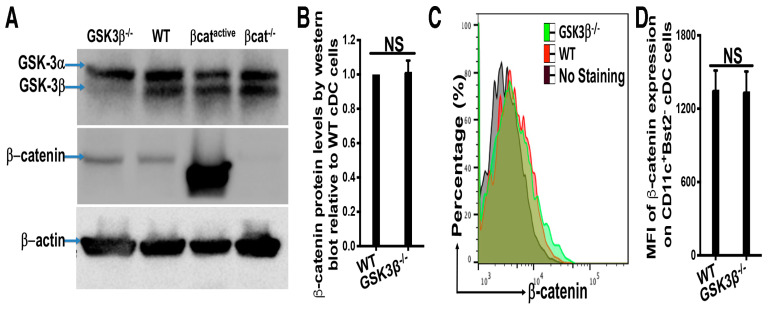
Deletion of GSK-3β in DCs does not upregulate β-catenin. (**A**,**B**) GSK-3β^−/−^ cDCs express similar levels of β-catenin to WT cDCs. WT and GSK-3β^−/−^ splenic cDCs were isolated and subjected to Western blot. (**A**) Expression of GSK-3α/β, β-catenin, and β-actin by Western blotting is shown. One of three experiments is shown. (**B**) Statistical analysis of β-catenin expression is shown. The relative expression of β-catenin Western blot intensity relative to that of b-actin loading control was calculated, and the ratios for WT cDCs for each experiment were set at 1.0. (**C**,**D**) Deletion of GSK-3β in DCs does not upregulate β-catenin. Histogram overlay of β-catenin expression (**C**) and mean fluorescence intensity (MFI) of β-catenin expression (**D**) on gated CD11c^+^Bst2^−^ cDCs are shown. Student’s *t*-test, and NS > 0.05. Data shown are representative of at least three experiments.

**Figure 4 vaccines-12-01037-f004:**
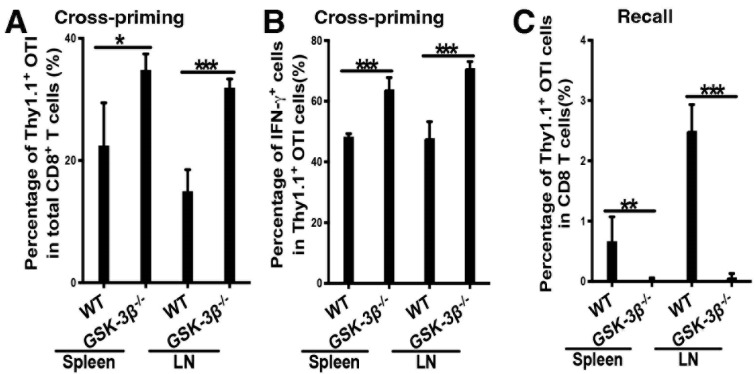
Deletion of GSK-3β in DCs abrogated memory CD8 T cell responses despite augmented cross-priming. (**A**,**B**) Deletion of GSK-3β in DCs led to significantly augmented cross-priming. WT and DC-GSK-3β^−/−^ mice (*n* = 4) were immunized with anti-DEC-205-OVA with CpG following adoptive transfer of naïve CFSE-labeled Thy1.1^+^ OTI cells, and cross-priming was examined at day 4 after immunization. (**A**) The percentages of Thy1.1^+^ OTI cells out of total CD8 T cells, and (**B**) the percentages of IFN-γ^+^ cells out of total Thy1.1^+^CD8^+^ OTI cells in both spleen and draining LN are shown. (**C**) CD8 memory responses were abrogated in CD11c-GSK-3β^−/−^ mice upon recall. Immunized WT and CD11c-GSK-3β^−/−^ mice (*n* = 4–5) were recalled at day 21 and examined 5 days later. The percentages of Thy1.1^+^ OTI cells out of total CD8 T cells are shown. Student’s *t*-test. * *p* < 0.05, ** *p* < 0.01, and *** *p* < 0.001. Data shown are representative of at least two experiments.

**Figure 5 vaccines-12-01037-f005:**
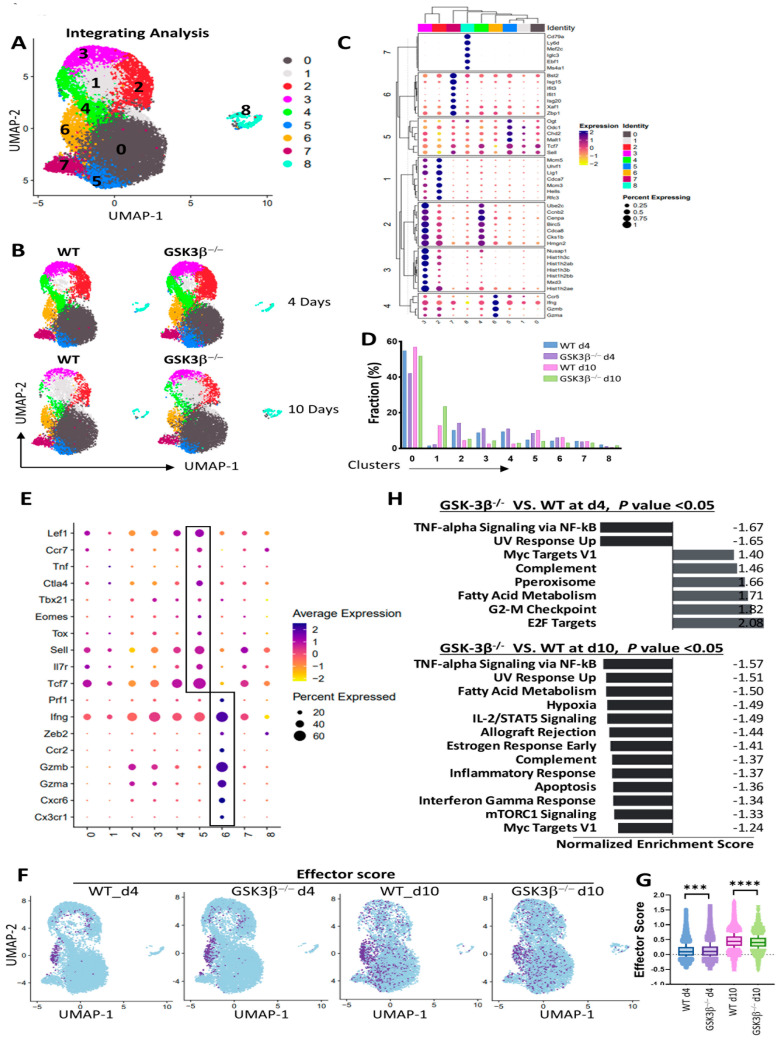
scRNA-seq of OVA-specific CD8 T cells identifies distinct populations and reveals differences between CD8 T cells primed in WT and CD11c-GSK-3β^−/−^ mice. WT and CD11c-GSK-3β^−/−^ mice adoptively transferred Thy1.1^+^ OTI CD8 T cells were immunized with anti-DEC-205-OVA plus CpG. Spleen cells were harvested at day 4 or day 10 after immunization, and FACS-sorted OTI cells were subjected to scRNA-seq as described. (**A**,**B**) UMAP-dimensionality reduction mapping analysis of single-cell gene expression data of OTI cells isolated 4 or 10 days following vaccination with ant-DEC-205-OVA. Each dot represents one single cell. A total of 9 clusters were identified and color-coded as indicated. UMAP visualization of single cells from combined OTI cells (**A**), or OT1 cells from WT or CD11c-GSK-3β^−/−^ mice at day 4 and day 10 (**B**) are shown. (**C**) Bubble plots depicting expression of top DEGs for UMAP clusters shown in (**A**). (**D**) Distribution of OTI cells from either WT or CD11c-GSK-3β^−/−^ mice at day 4 or day 10 within each of the 9 clusters as depicted in (**A**). (**E**) Bubble plots showing the key signatures for CD8 T cells effector and memory phenotype. (**F**) Expression of effector markers among the UMAP clusters. Gradient expression levels are color-coded as indicated. (**G**) Violin plot depicting the module score of gene sets associated with effector on OTI cells from either WT or CD11c-GSK-3β^−/−^ mice at day 4 or day 10. *** *p* < 0.001 and **** *p* < 0.0001 (**H**) Signaling pathways that are significantly downregulated or upregulated in OTI cells primed in CD11c-GSK-3β^−/−^ mice compared to OTI cells from WT mice at day 4 and day 10.

**Figure 6 vaccines-12-01037-f006:**
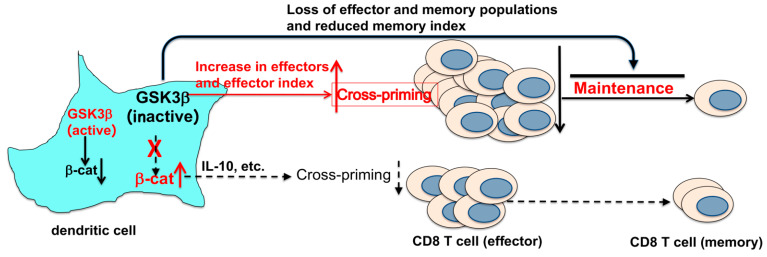
Schematic representation of GSK-3β’s dual roles in regulating CD8 T cell responses. Inhibition of GSK-3β is generally believed to upregulate β-catenin, leading to increased IL-10 production, which suppresses cross-priming and reduces memory CD8 T cell responses. However, our studies demonstrate that genetic deletion of GSK-3β in CD11c^+^ DCs does not result in β-catenin accumulation (activation). Instead, the deletion of GSK-3β in DCs enhances cross-priming of CD8 T cells, as indicated by an increase in effector cells and a higher effector index, based on scRNA-seq analysis. Despite this enhanced cross-priming, memory CD8 T cells are nearly abrogated in CD11c-GSK-3β^−/−^ mice, likely due to a significant loss of both effector and memory CD8 T cell populations. Collectively, these findings reveal novel mechanisms by which GSK-3β exerts opposing effects on CD8 T cell responses.

## Data Availability

The data that support the findings of this study are included in this manuscript and/or are available upon request by qualified researchers from the corresponding authors. scRNA-seq data have been deposited in GEO (accession number276602).

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
