# Peer review of "GSK-3β in Dendritic Cells Exerts Opposite Functions in Regulating Cross-Priming and Memory CD8 T Cell Responses Independent of β-Catenin"

_vaccines, 2024, doi:10.3390/vaccines12091037_

Round 1

Reviewer 1 Report

Comments and Suggestions for Authors

I would like to suggest to correct fig. 2 in section E with WT/GSK… in Third circle.

At Line 426 the “ratio” of sentence is right but it is not possible to consider that the approachs are different and they have to arrive to different results.

I woul add a figure that can explain  better the different results fronte cluster Lines after 120.

Author Response

(1)I would like to suggest to correct fig. 2 in section E with WT/GSK… in Third circle.”

Responses: We appreciate the insightful comments. We have modified Figure 2E to clearly mark the up or down is from either GSK3b-/- vs. WT  or b-cateninactive vs. WT. We have also revised the figure legends to make the description more clear.

(2) “At Line 426 the “ratio” of sentence is right but it is not possible to consider that the approachs are different and they have to arrive to different results.” 

Responses: We agree with the reviewer on the different approaches leading to different results. We have revised the manuscript to clearly state the different approaches used to block GSK-3b in the Discussion section.  

(3) “I would add a figure that can explain better the different results fronte cluster Lines after 120.” 

Responses: Thank you for your insightful comment. We have revised both the main text and the figure legend for Figure 2 to better clarify these clusters. Specifically, Figure 2B-F and Supplementary Figure 1 now provide a more detailed comparison of these clusters in GSK3b-/- vs. b-cateninactive DCs:

     Figure 2B displays the differentially expressed genes (DEGs) of key markers.

     Figure 2C highlights the top DEGs.

     Figure 2D illustrates the distribution of the 13 clusters.

     Supplementary Figure 1 shows the expression patterns of immune checkpoint

                      molecules, including Tim-3 and PD-L1/2, across these clusters.

     Figures 2E and 2F depict the relationship between up- and down-regulated genes for

                     GSK3b-/- vs. WT and b-cateninactive vs. WT.

ï‚·  Figure 2G presents the differentially regulated pathways.

Together, these Figures provide a comprehensive and detailed description of the difference between GSK3b-/- and b-cateninactive DCs.

Reviewer 2 Report

Comments and Suggestions for Authors

This paper reports new research on the role of GSK-3beta on the maturation of dendritic cells. The results lead to new insights. Rather than having effect via beta-catenin GSK-3beta has more complicated effects. On one hand it appears to stimulate cross priming and anti-tumor immunity, but inhibits memory CD8 T cell responses. This could have important implications for the use of GSK-3beta inhibitors in ongoing clinical trials, but this is not discussed. Also, in view of the rather complicated issues discussed, a cartoon depicting the the hypotheses about the mode of action of GSK-3beta before and after the study would have been helpful. 

Author Response

(1)This paper reports new research on the role of GSK-3beta on the maturation of dendritic cells. The results lead to new insights.

Responses: Thank you for your encouraging comments on the manuscript. 

(2) “Rather than having effect via beta-catenin GSK-3beta has more complicated effects. On one hand it appears to stimulate cross priming and anti-tumor immunity, but inhibits memory CD8 T cell responses. This could have important implications for the use of GSK-3beta inhibitors in ongoing clinical trials, but this is not discussed.” 

Responses: We appreciate the insightful comments and agree with the reviewer's point. We have now added detailed discussion of the potential implication of our findings on the targeting GSK-3 including GSK-3 inhibitors for cancer immunotherapies, in the Discussion section.

(3) “Also, in view of the rather complicated issues discussed, a cartoon depicting the hypotheses about the mode of action of GSK-3beta before and after the study would have been helpful.” 

Responses: We thank the reviewer for this excellent suggestion. We have now included a cartoon as Figure 6, which depicts how our studies provide new insight on how GSK-3b regulates DC vaccine-induced CD8 T cell responses.

Reviewer 3 Report

Comments and Suggestions for Authors

Using a gene-deleted mouse model, the authors clearly showed that GSK3B deletion in DC leads to oppositely directed immune responses: enhanced cross-priming (anti-tumor) and suppressed CD8 T memory responses (pro-tumor), independent of β catenin. The study performed a well-planned and excellent analysis, and the conclusion is reasonable based on the data.

Comments

Minor point.

  1. Their conclusion was not found regarding whether GSK3B should be deleted (or inhibited) or not to enhance the anti-tumor potential of the DC vaccine. Therefore, I think the message is not clear in the current MS. The authors are recommended to mention this in the Abstract and Discussion section.
  2. Maintenance of CD8T memory should be important for maintaining anti-tumor effects. However, the anti-tumor effect is enhanced in GSK3B deleted mice despite the loss of memory phenotype. The authors are recommended to discuss whether GSK3B should be suppressed clinically after all in the Discussion section.
  3. The authors are recommended to show the data on the presence of OT-I cells on day 21. If they do not exist on day 21, it is unclear whether the absence of cells after rechallenge indicates an inability to form memory T cells or a loss of memory function.

Author Response

(1) “Using a gene-deleted mouse model, the authors clearly showed that GSK3B deletion in DC leads to oppositely directed immune responses: enhanced cross-priming (anti-tumor) and suppressed CD8 T memory responses (pro-tumor), independent of β catenin. The study performed a well-planned and excellent analysis, and the conclusion is reasonable based on the data.

Responses: We thank the reviewer for your encouraging comments on the manuscript. 

(2) “Their conclusion was not found regarding whether GSK3B should be deleted (or inhibited) or not to enhance the anti-tumor potential of the DC vaccine. Therefore, I think the message is not clear in the current MS. The authors are recommended to mention this in the Abstract and Discussion section.

Maintenance of CD8T memory should be important for maintaining anti-tumor effects. However, the anti-tumor effect is enhanced in GSK3B deleted mice despite the loss of memory phenotype. The authors are recommended to discuss whether GSK3B should be suppressed clinically after all in the Discussion section.” 

Responses: We agree with the reviewer on these points. We have now added detailed discussion of the potential implication of our findings on the use of GSK-3 inhibitors for cancer immunotherapies, in the Discussion section. We have also stated the impact of our findings in the Abstract and Conclusion sections.

(3) “The authors are recommended to show the data on the presence of OT-I cells on day 21. If they do not exist on day 21, it is unclear whether the absence of cells after rechallenge indicates an inability to form memory T cells or a loss of memory function.” 

Responses: We would like to thank the reviewer for your insightful comments. We have observed that memory CD8 T cell responses were substantially reduced after recalled (challenged) at 21 days post immunization (Figure 4C). The complete loss of recalled CD8 T cell responses suggests that antigen-specific (OVA-specific) CD8 T cells were likely substantially reduced or lost at 21 days post immunization. Whether primed OTI cells failed to form memory OTI cells or lost their memory function is not clear. However, our scRNA-seq analysis suggests that both mechanisms are operational for the phenotype. We have added a paragraph discussion this important point in the Discussion section.